# DiQAD: A Benchmark Dataset for
# End-to-End Open-domain Dialogue Assessment

**Yukun Zhao**[1,2*]   **Lingyong Yan**[2*]   **Weiwei Sun**[1]   **Chong Meng**[2]
**Shuaiqiang Wang**[2]   **Zhicong Cheng**[2]   **Zhaochun Ren**[3†]   **Dawei Yin**[2†]
[1]Shandong University, Qingdao, China   [2]Baidu Inc., Beijing, China
[3]Leiden University, Leiden, The Netherlands
{zhaoyukun02,yanlingyong}@baidu.com,sunnweiwei@gmail.com
{mengchong01,wangshuaiqiang,chengzhicong01}@baidu.com
z.ren@liacs.leidenuniv.nl, yindawei@acm.org

## Abstract

Dialogue assessment plays a critical role in the development of open-domain dialogue systems. Existing work are uncapable of providing an end-to-end and human-epistemic assessment dataset, while they only provide sub-metrics like coherence or the dialogues are conversed between annotators far from real user settings. In this paper, we release a large-scale dialogue quality assessment dataset (DiQAD), for automatically assessing open-domain dialogue quality. Specifically, we (1) establish the assessment criteria based on the dimensions conforming to human judgements on dialogue qualities, and (2) annotate large-scale dialogues that conversed between real users based on these annotation criteria, which contains around 100,000 dialogues. We conduct several experiments and report the performances of the baselines as the benchmark on DiQAD. The dataset is openly accessible at https://github.com/yukunZhao/Dataset_Dialogue_quality_evaluation.

## 1 Introduction

Open-domain dialogue system (ODS) is quite popular in artificial intelligence (Serban et al., 2016; Huang et al., 2020; Bae et al., 2022), especially with the remarkable performance achieved in large language models (LLMs) (Wei et al., 2022; Ouyang et al., 2022; OpenAI, 2023). Dialogue assessment is critical to the development of open-domain dialogue systems (Deriu et al., 2021) as it guides what constitutes good conversational dialogues.

However, open-domain dialogue assessment still remains a challenging task (Deriu et al., 2021; Le et al., 2023). On the one hand, open-domain dialogue assessment is complicated and costly. A straightforward solution is the human-based method, i.e., recruiting human evaluators to interact with the dialogue systems and provide their

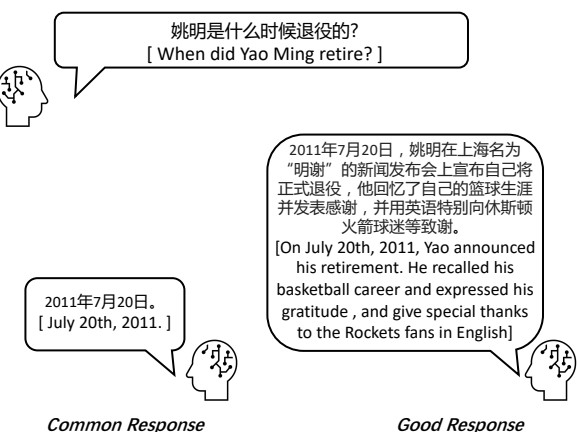

Figure 1: Both responses are satisfied, but the right one achieves higher quality due to its richer informativeness.

feedback (Kelly et al., 2009; Li et al., 2017b), which is often expensive and less reproducible. Other studies (Zhao et al., 2017; Gupta et al., 2019; Sai et al., 2020) release dialogues with references, which evaluate dialogue responses by comparing the generated responses with manually annotated references (Yang et al., 2015; Rajpurkar et al., 2016; Clark and Etzioni, 2016; Papineni et al., 2002). However, their performance is usually limited to reference coverage. Reference-free evaluation (Ghazarian et al., 2022; Sinha et al., 2020; Sai et al., 2020; Zhao et al., 2020) uses crafted samples to train evaluation models due to the lack of annotations, which only capture coarse-grained metrics like coherence.

On the other hand, most previous settings are less capable of end-to-end evaluation to determine whether the models perform like real humans. For example, the assessment datasets (Mehri and Eskénazi, 2020; Gopalakrishnan et al., 2019) focus on evaluating particular aspects of dialogues, e.g., coherence or diversity, while ignoring the overall quality. As shown in Figure 1, the previous settings consider both responses as satisfied ones and ignore their informativeness differences, lacking

---

*Contributed equally.
†Co-corresponding authors.

higher requirements like real humans. Furthermore, previously released datasets (Young et al., 2022; Komeili et al., 2022; Gopalakrishnan et al., 2019; Sun et al., 2021a) consist of dialogues conversed between recruited annotators and bots or between annotators, which varies from practice. As a consequence, the evaluating models learned from these datasets are far from optimal.

In this paper, we release a large-scale dataset - DiQAD (**Di**alogue **Q**uality **A**ssessment **D**ataset) for open-domain dialogue quality assessment, which aims to learn an atomically evaluating model to guide the dialogue system in producing both satisfied and high-quality responses. Specifically, we first establish unified human-epistemic quality criteria for open-domain dialogues based on 6 quality dimensions: grammaticality, relevance, consistency, empathy, proactivity, and informativeness (Finch and Choi, 2020). We set the quality label as a 3-scale holistic score (from 0 to 2) by considering the above dimensions, identifying higher quality dialogues that provide more comprehensive information and actively engage in the conversations using a score of 2. After that, we collect a substantial number of real dialogues from an online conversation platform and hire experienced annotators to annotate them following the above quality criteria. In total, **DiQAD** contains around 100k dialogues with around 0.8 million dialogue utterances across 6 common domains.

On the DiQAD, we conduct extensive experiments to verify the performance of different models. Additionally, the cross-domain experiments on DiQAD show that the best model learned on our dataset can generalize to unseen domain dialogues.

The contributions of this paper are twofold:

- This paper defines the rules for assessing dialogue quality and constructs a large-scale dialogue evaluation dataset. To the best of our knowledge, this is the first large-scale dataset focused on dialogue quality assessment.

- This paper conducts benchmark experiments on DiQAD and finds that recent large language models (e.g., ChatGPT, ChatGLM) are less capable of discriminating high-quality dialogues.

## 2 Related Work

**Dialogue System** Dialogue systems are quite popular, with the remarkable performance of large language models (Ouyang et al., 2022; OpenAI, 2023), and they are usually classified into task-oriented and open-domain dialogue systems (Ni et al., 2022). The task-oriented dialogues aim to solve specific tasks in a certain domain, which are supported by pipeline systems (Li et al., 2017a; Cheng et al., 2020; Williams et al., 2017; Golovanov et al., 2019) or end-to-end systems (Ni et al., 2022; Le et al., 2020; Wang et al., 2019; He et al., 2020). The open-domain dialogue system consists of chat-oriented systems aiming to converse with users without task and domain restrictions (Tao et al., 2019; Feng et al., 2020; Xu et al., 2020; Song et al., 2020; Miao et al., 2019; Wu et al., 2021), and conversational question-answering systems which are developed to answer specific questions (Liu et al., 2019; Sun et al., 2019; Qu et al., 2019).

**Dialogue Evaluation** There are two directions for dialogue evaluation: human evaluation and automatic evaluation. The human evaluation involves recruiting experts to test a dialogue system and then collecting questionnaires (i.e., user ratings) (Kelly et al., 2009; Ashwin et al., 2018; Li et al., 2017b). For the dialogues with predefined tasks, metrics like task-completion rate and task-completion cost are calculated from user interactions (Walker et al., 1997; Bodigutla et al., 2019) or user simulators (Schatzmann et al., 2007; Schatzmann and Young, 2009; Zhang and Balog, 2020; Sun et al., 2022) to evaluate the dialogues.

Another way for automatic evaluation is to evaluate the quality of dialogue contents One is reference-based evaluation, which measures the similarity between the generated responses and the ground-truth one, including correctness metrics such as MAP, MRR, EM, F1, and accuracy (Yang et al., 2015; Rajpurkar et al., 2016; Clark and Etzioni, 2016). Word-overlap statistics like BLEU, ROUGE, and METEOR (Papineni et al., 2002; Lin, 2004; Banerjee and Lavie, 2005), as well as neural metrics (Sato et al., 2020; Tao et al., 2018; Zhao et al., 2023), are calculated. However, it may suffer from the one-to-many problems (Zhao et al., 2017), even with multi-references (Gupta et al., 2019; Sai et al., 2020). Besides, recently released datasets are dialogues that occur between recruited annotators and bots (Young et al., 2022), or between annotators themselves (Wang et al., 2021; Smith et al., 2020; Komeili et al., 2022; Gopalakrishnan et al., 2019), or happen in certain scenarios like English practice (Li et al., 2017b). The conver-

sational topics and contents are handcrafted and limited compared to real users' settings.

The other one is reference-free evaluation, which train a classification model using users' ratings (Liang et al., 2020) or crafted samples (Ghazarian et al., 2022; Sinha et al., 2020; Sai et al., 2020; Zhao et al., 2020) for coarse-grained evaluation like coherence due to the short of annotated labels. Recent work (Mehri and Eskénazi, 2020; Deriu et al., 2020; Bao et al., 2019; Sun et al., 2021a; Le et al., 2023) use annotated samples to train the evaluation model. The annotations for open-domain dialogues are desirable, but the previous work only focuses on sub-type of dialogues such as task-oriented dialogues (Sun et al., 2021a). For open-domain dialogue annotation, they (Sun et al., 2021a; Bodigutla et al., 2019; Mehri and Eskénazi, 2020; Gopalakrishnan et al., 2019) evaluate some sub-metrics like coherence, diversity, empathetic or factuality, lacking of compound and human-epistemic high-quality evaluation and the data size is limited.

## 3 Open-domain Dialogue Quality Assessment

This paper treats open-domain dialogue quality assessment as a critical task, which differentiates higher quality dialogues based on real users' dialogues, to guide the future dialogue generation towards more satisfying and human-epistemic responses.

To this end, the quality assessment criteria are required to reflect higher quality for human cognition as much as possible. The human demands upon dialogue quality are usually regarded as entailing several fine-grained dimensions (Deriu et al., 2021; Smith et al., 2022). Inspired by a fully analyzed human-epistemic evaluation (Finch and Choi, 2020), we adopt the following 6 dimensions to establish our quality assessment criteria (see examples in Table 1):

- **Grammaticality**: Whether the utterances are fluent, readable, and free of grammatical and semantic errors.
- **Relevance**: Whether the responses logically match and are coherent with the questions.
- **Consistency**: Whether the utterances provide a consistent persona and no contradictions with the previously provided utterances.
- **Empathy**: Whether the respondent compre-

hends the user's feelings and appropriately reacts to emotional expressions (e.g., appeasing).
- **Proactivity**: Whether the respondent responds actively, provides useful extensions, and moves the dialogue to new topics.
- **Informativeness**: Whether the responses provide useful, specific, and sufficient information.

Based on the above dimensions, we conduct a 3-scale (0, 1, 2) quality assessment. A dialogue is set to at least answer the user's question; thus, the quality is at least a binary category of 1 and 0 to distinguish whether the user's question is resolved or not. Furthermore, we aim to identify high-quality dialogues that guide future dialogue generation towards higher quality, like humans. To achieve this, we introduce an additional category of 2 indicating a high-quality rating.[1]

The quality assessment includes turn-level and dialogue-level evaluation. We provide dialogue-level quality evaluation criteria. Specifically, we assess the quality of dialogue according to the following instructions (The examples are shown in Table 1, and detailed instructions and more examples are shown in Appendix A.2):

- **Low quality (score 0)**: The dialogue fails to meet the basic requirements, including grammaticality, relevance, consistency, and empathy. As shown in Table 1, the response of the first case is irrelevant to its question. Thus we label it 0-class directly without considering other metrics.
- **Moderate quality (score 1)**: The dialogue can basically answer the user's question, providing grammatical, relevant, consistent, and empathetic responses but fails to meet proactivity and informativeness. As the second case in Table 1, the response is only one reason that fails to supply sufficient information to meet the informative requirement.
- **High quality (score 2)**: The dialogue meets all the dimensions mentioned above. It is a higher requirement in addition to correctly answering questions on 1-class dialogue, which provides more comprehensive information and actively engages in conversations with

---

[1]Our annotators show lower annotation agreement under a more fine-grained, like 1-5 scale assessment for dialogue quality. We leave the 1-5 scale version for future work.

| Dialogue | Score | Explanation |
|---|---|---|
| **Q**: Any suggestions for buying iPhone 14 or 14 pro? 
 **R**: The iPhone is designed and marketed by Apple Inc. Apple is an American multinational technology company ... | 0 | Not relevant between buying and market |
| **Q**: My girlfriend is annoyed at my ex. 
 **R**: Perhaps you remain affiliated with your ex. When you break up with your ex completely, she would not be upset anymore. | 1 | Give only one reason that do not provide complete information |
| **Q**: Which products are manufactured by Foxconn? 
 **R**: Foxconn manufactures electronic products including the iPhone, MacBook, Dell, HP computers, and other consumer electronics. 
 **R**: Foxconn manufactures electronic products for major American, Canadian, Chinese, Finnish, and Japanese companies. It is a multinational electronics contract manufacturer ... | 2 | Provide the all desired information. |

Table 1: Dialogue quality assessment examples following our criteria. The Q and R denote the question and response respectively. All contents are translated from Chinese.

the user, as shown in the third case in Table 1 and Figure 9 in Appendix A.2.

In the next section, we introduce how we build the DiQAD based on the above criteria.

## 4 DiQAD: A Large-scale Dataset for Dialogue Quality Assessment

In this section, we describe how we construct the DiQAD based on the above annotation instructions in detail. We first collect amounts of real dialogues from an online platform. Then, we recruit experienced annotators to annotate these dialogues. In the last part of this section, we will show the detailed statistics of DiQAD.

### 4.1 Real Dialogue Collection

In the first step, we collect real dialogues in text from WenYiWen[2]. WenYiWen provides online conversational services, in which users can chat with others and obtain required information from experienced experts. Users can propose their questions to the platform, and then experienced experts can browse them and choose their interested ones to start a dialogue. After that, the dialogue takes place between the user and the expert until the user obtains their satisfied information or the duration reaches its limit. WenYiWen provides domain and topic tags for user questions to help experts select their related questions. The experts are invited from various professions, such as lawyers, university professors, college students, and car mechanics. Besides, to attract enough experts, users are asked to pay for their questions, and the experts can receive cash rewards based on the dialogue quality.

[2] https://wen.baidu.com

Specifically, we collect real dialogues in 6 domains from publicly accessible WenYiWen dialogues set[3]. The domains we selected consist of emotion, digital, legal, education, car, and finance dialogues. Consequently, the collected dialogues for annotation are more than 100 thousand (The statistics of collected dialogues are shown in Section 4.3).

### 4.2 Quality Annotation

Next, we recruit dozens of experienced annotators to annotate the dialogue quality on a crowdsourcing platform developed by ourselves.

The annotation process is an annotation-checking procedure, or repeated annotation-checking procedures if the annotation accuracy of the first round of annotation does not match the requirements. The dialogues are divided into multiple batches for annotation, with each batch consisting of 2000 samples. The annotators are divided into several groups, with each group annotating the batches of data.

For each dialogue, the quality label has been agreed upon and confirmed by 2 annotators. The annotator is first presented with the whole dialogue text. To guarantee the labeling reliability, the reading duration of each dialogue is required to be no less than a fixed time. After that, annotators are asked to label the quality score following the annotation criteria in section 3 and the detailed instructions in Appendix A.2. In addition to the quality labels, annotators are also asked to give explicit reasons why the labels are decided to improve annotation quality.

[3] For each WenYiWen dialogue, users are asked whether to open their dialogues to the public. If users agree to open, then the dialogue will become publicly accessible.

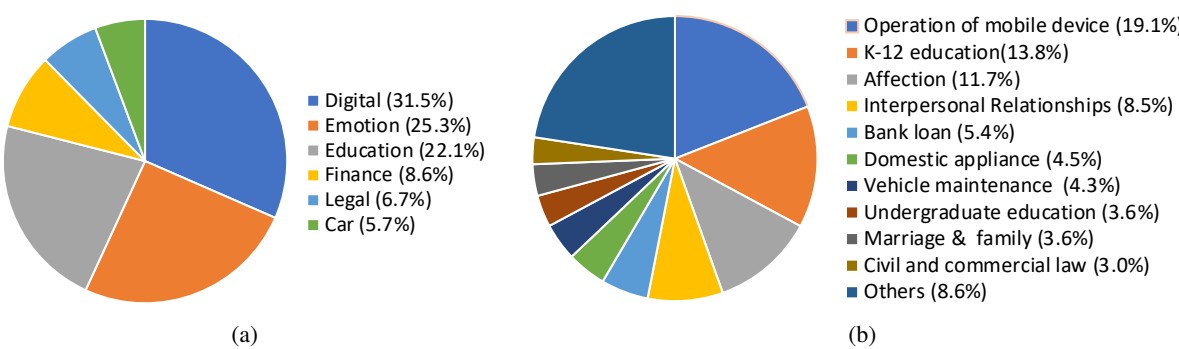

Figure 2: The domain (a) and topic (b) distributions of DiQAD.

After annotating a batch of data, we have a professional quality controller to check the label accuracy. (The roles and distribution of crowd-source annotators can be found in Appendix A.3). Our acceptance criterion is to achieve an annotation accuracy of 92% or above. More detailed information about annotation is shown in Appendix A.

### 4.3 Dataset Statistics

Finally, we obtain DiQAD, consisting of 100k dialogues across 6 different domains. Its basic statistics are shown in Table 2. DiQAD contains over 100k dialogues, where each dialogue consists of around 8 utterances and 334 tokens on average. In Table 2, we also compare our proposed DiQAD with recent popular open-domain datasets, DailyDialogue (Li et al., 2017b), BlendedSkillTalk (Smith et al., 2020), NaturalConv (Wang et al., 2021), FusedChat (Young et al., 2022), WizInt (Komeili et al., 2022), USR (Mehri and Eskénazi, 2020), FED-Dial (Mehri and Eskenazi, 2020), and TopicalChat (Gopalakrishnan et al., 2019), while the first 5 datasets have no quality annotation. NaturalConv and DiQAD are written in Chinese, and the others are in English. The scale of DiQAD is larger, and it is a compound human-epistemic evaluation in real users' dialogues compared with all other datasets.

The domain distributions of DiQAD are shown in Figure 2a. We see that the top 3 domains in DiQAD are digital, emotion, and education, with the smaller ones are finance, legal, and car. We utilize the built-in topic tags mentioned in section 4.1 and show the top 10 topics in DiQAD in Figure 2b. We see that DiQAD spreads over diverse topics such as the operation of mobile devices, affection, bank loans, etc., which can reflect the real users' settings to some extent.

| Dataset | #insts. | #uttrs. | #tokens | q-label |
|---|---|---|---|---|
| DailyDialogue | 13k | 7.9 | 114.7 | N |
| BlendedSkillTalk | 5k | 11.3 | 155.4 | N |
| NaturalConv | 19k | 20.1 | 244.8 | N |
| FusedChat | 10k | 5.8 | 65.2 | N |
| WizInt | 10k | 9.7 | 185.2 | N |
| USR | 660 | 9.3 | 180.2 | Y |
| FED-Dial | 125 | 12.7 | 113.8 | Y |
| TopicalChat | 10k | 21 | 399 | Y |
| DiQAD | 100k | 7.9 | 334.1 | Y |

Table 2: The statistics of DiQAD and other open-domain dialogue evaluation datasets. We list the number of dialogue instances, the mean number of utterances per dialogue, the mean number of tokens per dialogue, and whether containing quality labels.

## 5 Experiments

In this section, we conduct and illustrate benchmark experiments on DiQAD, including examining the overall performance of benchmark models (§5.2), conducting a detailed analysis based on the best benchmark model (§5.3), comparing to other reference-free evaluation methods (§5.4), and performing hyperparameter analysis (§5.5).

### 5.1 Experimental Setup

**Dataset.** We randomly split DiQAD into training, validation, and test sets, where the validation and test sets consist of 10,000 dialogues, and the remaining dialogues are used for training. For all models trained on the training set, we tune their parameters on the validation set and report the performance results on the test set. The final test set contains 1,958 samples classified as score 0, 3,628 samples classified as score 1, and 4,414 samples classified as score 0.

**Metrics.** We employ the following metrics of baseline models[4]: (i) *Acc*, the accuracy of the model predictions; (ii) *Unweighted average recall (UAR)*, the arithmetic average of class-wise recalls; (iii) *Cohen's Kappa coefficient (Kappa)* (Cohen, 1960), which we used to measure the agreement between model-predicted labels and human-annotated labels; (iv) *Spearman's rank correlation coefficient (Spearman)*, the non-parametric measure of the correlation between model predictions and human labels; (v) *Pearson correlation coefficient (Pearson)*, the measure of linear correlation between model predictions and human labels. Besides, we also report the precision, recall, and F1-score of model predictions.

**Benchmark Models.** We use the following two types of models as our benchmark models: classical methods, Transformer-based methods, and LLM-based methods.

- *Classical methods*: Naive Bayes (Bayes), Logistic Regression (LR), XGBoost, and GRU.

- *Transformer-based methods*: Vanilla Transformer (Vaswani et al., 2017), a 4-layer Transformer; BERT (Devlin et al., 2019); MENGZI (Zhang et al., 2021), an efficiently pre-trained Transformer using a 300G Chinese corpus; ERNIE (Sun et al., 2021b), a 12-layer Transformer encoder pre-trained on a 4TB corpus consisting.

- *LLM-based methods*: ChatGLM-6B (Du et al., 2022), an open-source Chinese LLM; and ChatGPT (Ouyang et al., 2022), an LLM developed by OpenAI and accessed through `gpt-3.5-turbo-0301` API.

For Transformer-based models, we use the same data format as the input. Specifically, given a dialogue, we use "`question:`" and "`response:`" to annotate the sentences from the questioner and respondent, respectively. We then splice the dialogue into a long text and feed it into the models to get predictions. For Transformer-based models, we add two special tokens, `[CLS]` and `[SEP]`, to the beginning and end positions after tokenization. After encoding the dialogue using Transformer-based models, the embedding of the `[CLS]` token is fed into a 3-class classification layer to get predictions.

---

[4]We use the implementation from SciPy (`https://scipy.org/`) to calculate the Spearman and Pearson scores.

For LLM-based methods, we use the instructions detailed in Appendix C for few-shot learning.

**Implementation.** The neural models are optimized using the cross-entropy loss. We set the batch size to 16, the learning rate to $1e-5$, use the AdamW optimizer to optimize parameters, employ gradient clipping with a maximum gradient norm of 1.0, train up to 10 epochs, and select the best checkpoints based on performance on the validation set. Under default conditions, we use the ERNIE-base model and set the maximum input length to 512 tokens. We conduct analytical experiments on hyperparameters such as model size and maximum length in Section 5.5.

## 5.2 Overall Performance

The results of benchmark models are shown in Table 3. We can see that: (1) All transformer-based methods achieve higher performance than classical methods. This verifies the advantages of the Transformer-based architecture in natural language processing. (2) Pre-trained models, especially the Ernie model, outperform other methods by a large margin. This is not surprising because pre-training is widely regarded as the ability to incorporate external knowledge into model parameters. (3) LLMs-based methods, since they are not sufficiently trained on the dialogue assessment task, still have a significant gap compared to the training-based approach. Further exploration is needed on how to better utilize LLMs for dialogue evaluation tasks.

Since the Ernie model achieves the best results on DiQAD, we use it as the backbone for detailed analysis in the next section.

## 5.3 Detailed Analysis

**Cross-domain Generalizability** Firstly, we analyze the cross-domain generalizability of the model trained on DiQAD. We conduct comparison experiments by removing training samples from a particular domain during training and testing on that domain. The experimental results are shown in Table 4. For each domain, we also report Ernie's performance on it, where Ernie is trained with all training samples.

From Table 4, we can see that the model can still achieve comparable performance on domains not appearing in the training set. For example, all accuracy scores drop by less than 6% when the domain samples are removed from the training set.

| | Acc | UAR | Kappa | Spearman | Pearson | Precision | Recall | F1 |
|---|---|---|---|---|---|---|---|---|
| *Classical methods* | | | | | | | | |
| Bayes | 52.67 | 28.53 | 24.03 | 22.64 | 24.52 | 48.35 | 47.55 | 47.56 |
| LR | 60.98 | 32.99 | 36.83 | 37.66 | 39.60 | 57.74 | 54.98 | 55.21 |
| XGBoost | 63.37 | 34.56 | 40.54 | 42.22 | 43.35 | 62.06 | 57.59 | 58.21 |
| GRU | 61.34 | 33.72 | 37.68 | 39.98 | 41.15 | 60.02 | 56.11 | 56.53 |
| *Transformer-based methods* | | | | | | | | |
| Transformer | 64.16 | 35.02 | 41.61 | 45.52 | 46.66 | 63.12 | 58.62 | 59.28 |
| BERT | 68.23 | 38.76 | 48.91 | 55.59 | 55.37 | 67.92 | 64.61 | 65.71 |
| MENGZI | 69.25 | 39.79 | 50.91 | 56.28 | 56.29 | 68.28 | 66.31 | 67.07 |
| ERNIE | **69.94** | **40.96** | **52.55** | **57.39** | **57.25** | **68.68** | **68.27** | **68.46** |
| *LLM-based methods* | | | | | | | | |
| ChatGLM-6B | 36.70 | 21.08 | 3.05 | 9.44 | 10.09 | 36.22 | 36.70 | 34.22 |
| ChatGPT | 39.30 | 21.88 | 4.70 | 12.83 | 12.51 | 37.61 | 36.47 | 36.28 |
| ChatGPT + CoT | 41.00 | 24.91 | 10.68 | 17.79 | 17.33 | 30.77 | 31.14 | 30.38 |

Table 3: Comparison of benchmark models on DiQAD. The best results of each metric are shown in bold.

| | Acc | UAR | Spearman | Pearson | F1 |
|---|---|---|---|---|---|
| *Digital domain* | | | | | |
| ERNIE | 63.89 | 38.40 | 50.14 | 50.08 | 63.90 |
| - w/o Digital | 61.97 | 37.01 | 46.75 | 45.90 | 61.72 |
| *Emotion domain* | | | | | |
| ERNIE | 77.70 | 40.49 | 61.14 | 61.48 | 68.95 |
| - w/o Emotion | 72.14 | 38.74 | 56.74 | 56.97 | 64.25 |
| *Education domain* | | | | | |
| ERNIE | 69.82 | 40.55 | 52.13 | 54.78 | 68.06 |
| - w/o Education | 64.26 | 37.21 | 45.13 | 44.76 | 62.54 |
| *Legal domain* | | | | | |
| ERNIE | 72.63 | 41.00 | 56.73 | 55.71 | 69.27 |
| - w/o Legal | 72.00 | 40.23 | 59.94 | 58.65 | 68.41 |
| *Car domain* | | | | | |
| ERNIE | 67.71 | 38.56 | 47.93 | 47.03 | 64.49 |
| - w/o Car | 66.67 | 38.84 | 48.92 | 48.06 | 64.32 |
| *Finance domain* | | | | | |
| ERNIE | 68.35 | 38.96 | 53.61 | 52.77 | 65.80 |
| - w/o Finance | 67.41 | 37.75 | 51.30 | 50.33 | 64.43 |

Table 4: Cross-domain evaluation results. Each group in the table represents the results of a specific domain.

We also find that the metric decline rates are related to the number of samples in that domain. For example, the performance decrease of the model is more significant when a large domain (e.g., Emotion) is removed than a small domain (e.g., Finance).

**User Sentiment Influence**    To analyze the influence of user sentiment expression on dialogue quality assessment, we design two types of sentiment expressions and add them to the end of the original dialogues: (i) Positive sentiment: We use the template "*question: ok thanks! response: you are welcome!*"[5]. We denote the modified model as **ERNIE+pos**. (ii) Negative sentiment: We add

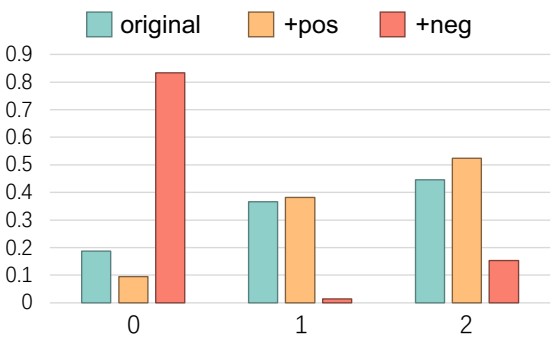

Figure 3: Comparison of predicted score distributions when adding different sentiment expressions.

the template "*question: I don't think so.*"[6]. We denote the modified model as **ERNIE+neg**. The experimental results are shown in Table 5. We also demonstrate the prediction score changes when adding different sentiment expressions in Figure 3.

From Figure 3 and Table 5, we can see that the model is more likely to predict a low-quality label when the dialogue contains negative user expressions and tends to give a higher quality if there is positive user feedback. As a result, the performance of **ERNIE+pos** decreases slightly compared to the original Ernie (e.g., the accuracy score drops by around 3and adding the negative expressions (i.e., **ERNIE+neg**) makes the accuracy score drastically decline by more than 38These performance changes, especially on the negative sentiment, show that quality assessment in DiQAD is sensitive to sentiment expressions. We believe that the sensitivity to user sentiment can help dialogue systems evolve in a direction that better satisfies the user.

---

[5]In Chinese: "*question:* 好的多谢！ *response:* 不谢！"

[6]In Chinese: "*question:* 不是这样的。"

|         | Acc   | UAR   | Spearman | Pearson | F1    |
|---------|-------|-------|----------|---------|-------|
| ERNIE   | 69.94 | 40.96 | 57.39    | 57.25   | 68.46 |
| ERNIE+pos | 66.99 | 36.63 | 50.60  | 51.12   | 62.51 |
| ERNIE+neg | 31.33 | 24.55 | 27.07  | 28.05   | 27.04 |

Table 5: Comparison of evaluation results when adding different sentiment expressions.

|                              | Spearman | Pearson |
|------------------------------|----------|---------|
| *Unsupervised methods*       |          |         |
| USR (Mehri and Eskénazi, 2020) | 4.47   | 11.64   |
|   - USR-MLM        | 2.61     | 3.83    |
|   - USR-DR         | 3.83     | 11.61   |
| USL-H (Phy et al., 2020)     | 4.60     | 7.91    |
|   - USL-VUP        | 0.65     | -1.22   |
| BARTScore (Yuan et al., 2021) | 24.73   | 32.77   |
| *Supervised methods*         |          |         |
| BLEURT (Sellam et al., 2020) | 48.37    | 49.96   |
| P-Tuning (Liu et al., 2021)  | 53.03    | 53.02   |
| Prompted ERNIE (3-shot)      | 15.09    | 28.34   |
|   - finetuning     | 56.49    | 56.49   |
| **ERNIE**                    | **57.39** | **57.25** |

Table 6: Results of reference-free dialogue evaluation methods. The best results are shown in bold.

## 5.4 Comparison with Other Reference-free Assessment Methods

In this section, we compare the performances of other reference-free assessment methods on DiQAD. There are usually two types of reference-free dialogue assessment methods: unsupervised methods and supervised methods. Unsupervised methods are usually learned without annotated dialogues, including: USR (Mehri and Eskénazi, 2020), USL-H (Phy et al., 2020), BARTScore (Yuan et al., 2021). Supervised methods are usually learned upon annotated dialogues, including: BLEURT (Sellam et al., 2020), P-Tuning (Liu et al., 2021), Prompted ERNIE (3-shot (Sun et al., 2021b).

The evaluation metric is *Spearman* and *Pearson*, following previous studies (Mehri and Eskénazi, 2020). The results are listed in Table 6. We see that the unsupervised methods perform unfavorably. For example, USR and USL-H perform well on previous dialogue evaluation benchmarks but show low consistency with human assessment on DiQAD. The main reason may be that in DiQAD, the user's questions are more difficult and the system needs to provide comprehensive responses to meet the user's needs, and simple metrics such as relevance and fluency are not well suited.

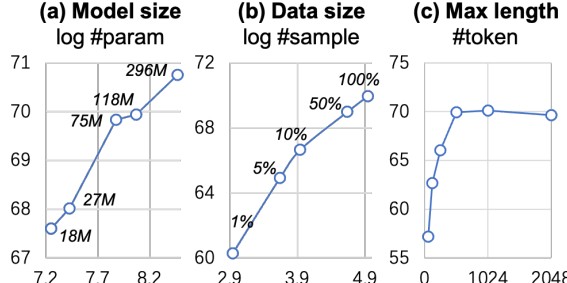

Figure 4: Hyperparameter analysis: (a) Model size, we demonstrate the Ernie performance with different numbers of parameters; (b) Data size, we show the effects of the training data size on the Ernie performance; (c) Max length, we compare the Ernie performances with different maximum input lengths.

## 5.5 Hyperparameter Analysis

We conduct hyperparameter analysis, and the results are illustrated in Figure 4. The aspect we analyzed and the findings are: (a) **Model size.** We use ERNIE of different sizes (e.g., 18M, 27M) and the results show that larger models achieve better performance. (b) **Data size.** We random sample a portion of training data (e.g., 1%, 5%) and train the ERNIE-base model. We find that more data lead to better performance and the increase curve is basically in line with the logarithmic speed. (c) **Input length.** We change the maximize input token number and test the performance difference. We find that increasing the maximum length can improve the effect when the length is less than 512, and there is almost no improvement after the length exceeds 512. This may be mainly because dialogue tokens in DiQAD are mostly less than 512.

## 6 Conclusion

In this paper, we conduct a human-epistemic dialogue quality assessment on real user conversations. We release a large-scale dataset DiQAD for open-domain dialogue quality assessment, which contains around 100k dialogues and 0.8 million utterances in 6 domains. We conduct several benchmark experiments on DiQAD to study the performances of classical, Transformer-based models and large language models on the dialogue quality assessment task. We hope the release of this dataset can inspire researchers in the community who lack such real and large-scale datasets, to evaluate real user dialogues to guide further improvement for the language models' generation.

## Limitations

The dialogue contents are written in Chinese, and hence the dataset can only be used to evaluate Chinese models. The impact may be restricted by its language. Besides, the benchmark models are absent from fine-tuning larger pre-trained models such as ChatGLM (Du et al., 2022) and Vicuna (Chiang et al., 2023). We cannot promise the current benchmark provide the strongest baseline. We believe that larger models and more fine-grained tuning would achieve better performance. Finally, the quality annotation has been simplified to 0-2 scale for higher labeling accuracy. More degree of labels may help to differentiate the dialogue quality more elaborately. We will study the above limitations in our future work.

## Ethics Statement

We acknowledge the importance of the ACM code of Ethics and totally agree with it. We ensure that this work is compatible with the provided code, specifically in terms of providing the dialogue dataset. We use the dataset in accordance with copyright terms and under the licenses of its provider.

**Licensing** We collect the dialogues that have been licensed from both the users and the respondent. All the dialogues have been granted to publish by their owner, i.e., the WenYiWen platform.

**Personal Information** We collect dialogue texts without user information. The collecting procedure obeys the privacy policy according to local laws.

**Toxic&Privacy** All the dialogues have been examined by the WenYiWen audit team one by one, to avoid pornography, offensiveness, profanity, privacy, and toxicity risks.

## Acknowledgements

This work was supported by the National Key R&D Program of China with grant No.2020YFB1406704, the Natural Science Foundation of China (62272274, 61902219, 61972234), the Natural Science Foundation of Shandong Province (ZR2021QF129). We thank the WenYiWen platform for providing the original dialogues and helping us to filter privacy and toxic dialogues.

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

# A  Annotation Details

## A.1  Annotation Process

We randomly sample the dialogues from WenYiWen platform that are licensed to be publicly available. We proceed to sample dialogues from six domains: emotion, digital, legal, education, car, and finance. After collect the dialogues from WenYiWen platform, we employee crowd-sourcing workers to annotate the quality labels for the dialogues (0, 1, 2).

The annotation process is an annotation-checking procedure, or repeated annotation-checking procedures (if the annotation accuracy of the first round of annotation does not meet the needs). The dialogues are divided into multiple batches for annotation, with each batch consisting of 2000 samples.

For each dialogue, the quality label have been agreed upon and confirmed by 2 crowd-source workers. After annotate a batch of data, we have a professional quality controller to check the the label accuracy. (The roles and distribution of crowd-source workers can be found in section A.3). Generally, a quality controller randomly samples 100 dialogues and check the label accuracy of this batch including overall annotation accuracy and the accuracy of each category (0, 1, 2). Our acceptance criterion is to achieve an overall accuracy of 92% above, and the accuracy for each category no less than 90% for each batch of data.

If the accuracy does not meet the thresholds, the entire batch of data is returned, and the existing labels are removed for re-annotation. This annotation and checking process is then repeated until the labeling accuracies surpass the thresholds. Based on the statistics of our annotation, approximately 31% of batches require re-annotation, which means these data need two or more rounds of annotation.

It is worth noting that we continually annotate the dialogue quality data until now. And we sample 100k annotated dialogues for the open-domain dialogue quality assessment task for the community. We have applied the annotation in actual usage and were able to demonstrate real improvement when we use the data to train a dialogue quality model to determine the quality of real dialogues. The user satisfaction and user revisit ratio in WenYiWen platform have been improved 56%, 23% relatively in the last year after deploying such dialogue quality evaluation.

## A.2  Annotation Instructions

For each annotator, we provide instructions to enable them to perform the dialogue quality assessment task, as shown in Figure 5.

Each annotator take the form of a dialogue evaluation as shown in the following Figure 6. This task includes to answer whether the information provided need to be verified through external knowledge in case of hallucinating problems, rate the quality score from -2, -1, 0, 1 to 2 and choose the reason why you rate the score. The -1 and -2 are rated when there are privacies and personal information contained in the responses and questions respectively. We use annotators to avoid the ethics. We exclude the dialogues that score -2 and -1 for the consideration of ethics. This is the interface translated from Chinese, and the original Chinese interface is shown in Figure 10.

The detailed guidelines for scoring 0, 1, and 2 are shown in the following Figure 7, Figure 8, Figure 9 respectively.

---

In this task, you will evaluate the quality of a dialogue provided here.
To correctly solve this task, follow these steps:
    1. Read the dialogue carefully, be aware of the information in the questions and responses.
    2. Decide whether the information supplied need to be verified through external knowledge.
    3. Rate the dialogue quality from 0 to 2 scale by its grammaticality, relevance, empathy, proactivity and informativeness.
    4. Choose the reasons why you score 0, 1 or 2.

Figure 5: The task instruction used for dialogue quality evaluation.

> **Dialogue:**
> Q1 : How to retrieve a forgotten bank card number?
> A1: Hi. You can only retrieve it at the bank counter for the customer service. The bank card number refers to the code issued by various banks, and there are unified regulations for the numbers between different banks. The 6 digits of a credit card number represent the issuing bank identification code, also known as the BIN (Bank Identification Number). Different BIN numbers symbolize different bank institutions and card levels. The central bank uniformly assigns business number ranges to commercial banks, and each bank has different number ranges. Once a bank card is assigned, the customer's basic information, credit limit, and transaction status are promptly transmitted to the central bank to prevent money laundering and facilitate data aggregation and unified management in the central bank's database.
> A2: Hello?
> Q2: I have forgotten my ICBC (Industrial and Commercial Bank of China Limited) card number.
> A3: You still need to go to the according bank counter for help.
>
> ➤ **Is the information provided in the answers need to be verified through external knowledge?**
>    ⚪ 1        ⚪ 0
> ➤ **Rate the quality score.**
>    ⚪ 2      ⚪ 1      ⚪ 0      ⚪ -1      ⚪ -2      ⚪ Abandon-for-phone-number
> ➤ **Choose the reason why you score 1.**
>    ⚪ Incomplete        ⚪ Less informative      ⚪ Less proactivity      ⚪ None
> ➤ **Choose the reason why you score 0.**
>    ⚪ No/wrong answers ⚪ Not relevant      ⚪ Not empathy      ⚪ Grammatical Erro
>    ⚪ Not consistent      ⚪ None

Figure 6: The interface for annotators to label the dialogue quality. This is a translated version from Chinese.

## A.3 Annotators

The annotators are full-time crowdsourcing annotators who continuously annotate the dialogues to serve the commercial WenYiWen platform. We sample 100k dialogues to be published to the community for the dialogue quality evaluation task.

We maintain an annotation team of approximately 26 employees, including 2 trainers, 4 quality-controllers, and 20 annotators. All of them have at least a colledge degree. The 20 annotators carry out specific annotations, while the quality-controllers randomly review 100 annotations for each batch data after it has been annotated, to assess the annotatation accuracy against demands. Trainers are responsible for collecting cases of inconsistent annotations, discussing and formulating the annotation metrics with us and training the reviewers and annotators using the annotation criteria.

## B Baseline Details

Unsupervised methods are usually learned without annotated dialogues, which include:

- **USR** (Mehri and Eskénazi, 2020), which is a holistic score for quality evaluation using the RoBERTa as the backbone. USR is composed of two sub-scores: USR-MLM and USR-DR, where USR-MLM is calculated using the log-likelihood values of responses, and USR-DR finetunes the model with randomly sampled negatives.

- **USL-H** (Phy et al., 2020) is composed of VUP, NUP, and MLM, in which NUP and MLM are basically the same as USR-DR and USR-MLM, respectively, and VUP finetunes the model with synthetic negatives.

- **BARTScore** (Yuan et al., 2021), which uses the log-probability of BART generating answers conditioned on questions as the score.

Supervised methods are usually learned upon annotated dialogues, including:

| Dimension | Explanation | Case Study |
|---|---|---|
| No Answer | Do not provide answers | |
| Wrong Answers | Provide the wrong answers through the annotators' verification. | Q: How many prefecture-level cities are there in Jiangxi Province?
A: About 23.
(Inconsistent with ground-truth answer 11, a wrong answer) |
| Not relevant | The responses are not relevant with the questions. | Q: How many Universities in Zhuhai?
A: Zhuhai is a city in Guangdong province, and is 140 kilometres (87 miles) southwest of Guangzhou. |
| Not empathetic | The responses are offensive, or don't appropriately reacts to emotional expressions. | Q: I am sick and went to the Zhejiang hospital, but I have to wait for a notification for admission. Can I wait until then?
A: Ask your doctor directly. You don't even know how to get to the hospital as an adult? |
| Not consistent | Provide the inconsistent answers between different round of answers. | Q: An HP printer runs out of paper and cannot print even after the paper is loaded, how should it be resolved?
A: The sensor did not detect the presence of paper. You need to restart the power.
A: Do not turn off the power … |
| Not Grammatical | The responses are not fluent, readable, and exists grammatical and semantic errors. | Q: My classmates in the always tease me.
A: When someone likes you, it will to get your attention or intentionally pick on you. It's actually a way of expressing your feelings. That's the sentence meaning. |

Figure 7: The detailed guidelines to rate 0 for the quality of a dialogue.

| Dimension | Explanation | Case Study |
|---|---|---|
| Not complete - Informativeness | Not sufficient. Provide the basic answers, but not complete information | Q: The characteristics of winter in Perth, Australia.
A: Perth has a Mediterranean climate, with wet winters and long, hot summers. The hot weather during summer is caused by heatwaves blowing in from the eastern desert. |
| Not useful - Informativeness | Not specific and useful. Only give a direct answer to the question, and don't provide any related extensions for helping understanding the answers. | Q: When the car dashboard displays an icon of a small car with a key inside, what does it mean?
A: That is the indicator light for the car's anti-theft system.
(Absence of explanation, why the system is active and how to solve it ) |
| Not proactive | When the question is a single noun or a relatively simple statement without a clear demand, the respondent don't further explore the user's needs but provides answers directly. | Q: Tsinghua University
A: It is a nationally renowned key university. It is a member of the C9 League, the Song League, the Association of Asian Universities, the Association of Pacific Rim Universities and the Tsinghua-Cambridge-MIT Low Carbon University Alliance. … |

Figure 8: The detailed guidelines to rate 1 for the quality of a dialogue.

| Dimension | Explanation | Case Study |
|---|---|---|
| All | The question has been fully addressed. It is an informative answer that meets the criteria of relevance, consistency, grammaticality, and empathy. | Q: Let someone do or let someone to do in English?
A: "let someone do" is more commonly used.
A: The verb "let" is a causative verb. After take an object, it is followed by the base form of the verb. |
| All | The answer completely addresses the emotional feelings of the questioner and provides a proactive and comprehensive communication. And meets the criteria of relevance, consistency, grammaticality, and empathy. | Q: How to console someone when their loved one is sick?.
A: Can you give any help? At this time, when you can't be of direct help, you can send a message of comfort like "Everything will be fine. Let me know if there's anything I can do to help."
Q: Just send that message?
A: You can also send "Don't worry, good fortune will come to your family. Take care of yourself as well."
Q: Can I send this?
A: It's even better and more thoughtful.
A: Just send it, don't hesitate too much.
A: Don't worry too much either.
Q: Okay. Thank you? |

Figure 9: The detailed guidelines to rate 2 for the quality of a dialogue.

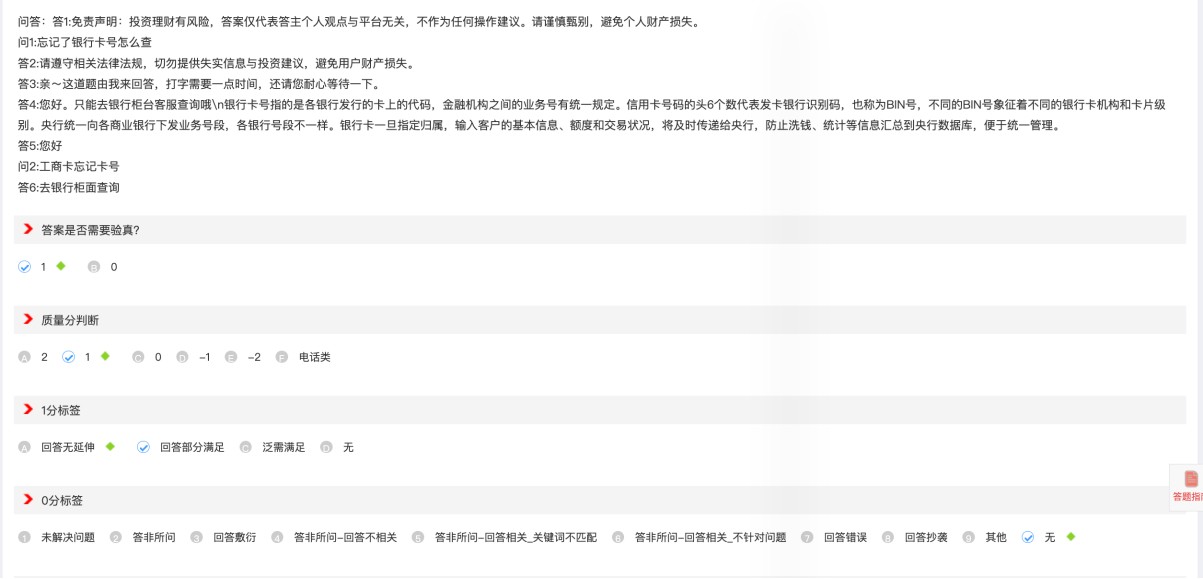

Figure 10: The interface for annotators to label the dialogue quality in Chinese.

- **BLEURT** (Sellam et al., 2020), in which a BERT-large model is pre-trained on synthetic data and fine-tuned with regression loss;

- **P-Tuning** (Liu et al., 2021), which optimizes the parameters of the prompt vector and freezes other parameters of ERNIE;

- **Prompted ERNIE (3-shot)** (Sun et al., 2021b), which uses 3 annotated dialogues as in-context examples to prompt an ERNIE model; we also test supervised finetuning setting by finetuning the parameters of the prompted model on the training set.

## C Prompts

**Chinese prompt**

**USER**
我会给出来自user和assistant的对话。对话以json格式保存，每项包括了其角色role（user或assistant）和消息内容content。请根据assistant的表现对其回复质量打分，标准如下：
- 0分，低质量
- 1分，中等质量
- 2分，高质量
注意你只需要输出一个分数代表回复的质量，不需要做任何解释。

**ASSISTANT**
好的，请提供对话，我将给出0到2的分数。

**USER**
['role': 'user', 'content': '不超过十字的新闻。快！快！', 'role': 'assistant', 'content': '再短的新闻也很难低于40个字。不超过十字的就是新闻标题，如下香港机场航班大致正常，少量取消部分延误天津一中学生遭班主任体罚横纹肌溶解金融委全面加大对实体经济的支持力度国庆假期火车票明天开抢']

**ASSISTANT**
分数：0

**USER**
['role': 'user', 'content': '平安保险生效7天2万还能退多少', 'role': 'assistant', 'content': '您好，很感谢您的提问，可以全额退款，因为生效7天还属于犹豫期，犹豫期退保是全额退的。希望我的回答能够帮助到您，祝您生活愉快！']

**ASSISTANT**
分数：1

**USER**
['role': 'user', 'content': '那去哪里可以查的到自己有没有被起诉呢', 'role': 'assistant', 'content': '通常去调取信息的话，您可以选择去公安局或是法院，提供您和您要查询的案件或当事人的身份关系，另外还有您的身份证以及户口本，还有就是说明您要查询的原因（这个法院那边会让您填表或是给您直接进行登记的）。您可以带好这些材料，法院那边都是可以帮您查的。']

**ASSISTANT**
分数：2

**USER**
{{dialogue}}

## English prompt (translated version)

**USER**

I will provide you a dialogue between user and assistant. The dialogue is saved in json format, in which each item include the role (user or assistant) and the message content. please rate the assistant performance based on the response quality, and the criterion is as follow:
- Score 0, low quality.
- Score 1, moderate quality.
- Score 2, high quality.
Note that you only need to output the number of the score, without any explanation

**ASSISTANT**

OK, please provide the dialogue and I will rate it in scale of 0-2.

**USER**

['role': 'user', 'content': 'News that does not exceed the cross. Hurry up! Hurry up!', 'role': 'assistant', 'content': 'Even the shortest news can hardly be less than 40 words. No more than the cross is the Headline, as shown below, flights at the Hong Kong airport are generally normal, a small number of cancellations and partial delays, a middle school student in Gamma Cygni was physically punished by the head teacher, and the Striated muscle tissue was dissolved. The Financial Committee fully increased its support for the real economy. Train tickets will be robbed tomorrow during the National Day holiday']

**ASSISTANT**

Score: 0

**USER**

['role': 'user', 'content': 'How much can I refund 20000 yuan after Ping An Insurance takes effect for 7 days', 'role': 'assistant', 'content': 'Hello, thank you very much for your question. We can offer a full refund as the 7-day validity period is still a hesitation period, and refunds during the hesitation period are full refunds. I hope my answer can help you, and I wish you a happy life!']

**ASSISTANT**

Score: 1

**USER**

['role': 'user', 'content': 'Where can I find out if I have been prosecuted', 'role': 'assistant', 'content': 'If you usually need to retrieve information, you can choose to go to the public security bureau or court to provide your identity and relationship with the case or party you want to inquire about, as well as your ID card and household registration book, and explain the reason for your inquiry (the court will ask you to fill out a form or directly register it for you). You can bring these materials with you, and the court can help you investigate them.']

**ASSISTANT**

Score: 2

**USER**

{{dialogue}}