# OpenReview forum: "DiQAD: A Benchmark Dataset for Open-domain Dialogue Quality Assessment"
_EMNLP/2023/Conference — EMNLP 2023 Findings_

### Official Review · Reviewer_NmNL · 2023-08-02

**Soundness:** 4

**Excitement:**

3: Ambivalent: It has merits (e.g., it reports state-of-the-art results, the idea is nice), but there are key weaknesses (e.g., it describes incremental work), and it can significantly benefit from another round of revision. However, I won't object to accepting it if my co-reviewers champion it.

**Paper Topic And Main Contributions:**

This paper proposes a large-scale Chinese dataset for dialogue evaluation called DiQAD containing 100k dialogues. Different from the previous work, they focused on human-centered evaluation with proposed six aspects: grammaticality, relevance, consistency, empathy, proactivity, and informativeness. And the dialogues are directly crawled from real-world scenarios instead of created by the annotators. After collecting the original dialogues, their annotation team annotates the dialogue-level quality in the six aspects. Since there are two annotator trainers and four quality controllers in the team, the quality of annotations can also be guaranteed. After constructing the dataset, the authors also conduct extensive experiments to evaluate various comparison methods for classifying the dialogue quality. This benchmark can also be considered as a contribution for future researchers.

**Questions For The Authors:**

+ [Solved] Question A: In Lines 375-378, you wrote '1,958 samples classified as score 0, ..., and 4,414 samples classified as score 0'. It must be a fault and what exactly the numbers of samples scored as 0, 1, and 2?
+ [Solved] Question B: In Table 4, why did the UAR, Spearman correlation, and Pearson correlation drop in legal and car domains when using these in-domain data?

**Reasons To Accept:**

1. The proposed large-scale annotated dataset for dialogue evaluation is certainly a good contribution to the NLP community.
2. Empirical studies are extensive, involving various comparison methods and diverse evaluation metrics. The experimental analysis also provides some valuable conclusions for this field.

**Reasons To Reject:**

1. The proposed aspects of human-centered evaluation are not novel. USL-H (Phy et al., 2020) has already considered Grammaticality, Relevance, Consistency, and Empathy. Informativeness is also a well-studied aspect in the field of knowledge-grounded dialogue systems. There are also many discussions on proactive dialogue systems (Liao et al., Proactive conversational agents, WSDM 2023 tutorial).

2. The length of dialogues in the dataset is short (less than 512 tokens on average), which may not be suitable for the real-world scenario that requires long-term dependency.

**Reproducibility:**

4: Could mostly reproduce the results, but there may be some variation because of sample variance or minor variations in their interpretation of the protocol or method.

**Reviewer Confidence:**

4: Quite sure. I tried to check the important points carefully. It's unlikely, though conceivable, that I missed something that should affect my ratings.

**Typos Grammar Style And Presentation Improvements:**

### Typos
+ Line 394: two types -> three types
+ Line 956: colledge -> college

### Grammar
+ Line 540: random -> randomly

---

> ### Author Rebuttal · Authors · 2023-08-29
>
> We sincerely thank you for your valuable feedback. The followings are our responses to your concerns in order.
>
> **Our Contribution**: The primary purpose of our paper is to provide human-epistemic assessments of the open-domain dialogues in real users settings, rather than providing new metrics. Recently released datasets comprise dialogues that are conversed between recruited annotators and bots (e.g., FusedChat (Young et al., 2022)) or between annotators themselves (e.g., NaturalConv (Wang et al., 2021), TopicalChat (Gopalakrishnan et al., 2019), BlendedSkillTalk (Smith et al., 2020), WizInt (Komeili et al., 2022)), or happened in certain scenarios like English practice (e.g., DailyDialogue (Li et al., 2017b)). The conversational topics and contents are handcrafted and limited compared with the real users’ dialogues, making them less suitable for the real user dialogue evaluation. In contrast, we present a dataset of real user-to-human dialogues with human-epistemic quality labels.
>
> The different lengths demonstrate that the dialogues in open assistants and dialogues in real-user scenarios do exhibit differences. The responses generated by open assistants have a distinct machine style, towards longer responses to enhance interactions with users. However, in real user dialogues, the goal should be to have as few turns as possible to meet the user's needs. With the rapid improvement of open assistants, our subsequent work will sample chinese dialogues from the known open assistants and annotate them using the existing process, thereby expanding the diversity of the dataset.
>
> **Response for Question A**: The testset contains 10,000 samples, among 1,958 of them are scored 0, 3628 samples are scored 1 and the reset 4,414 samples are scored 2. Thank you for pointing out this writing error and we will correct it in the next revision.
>
> **Response for Question B**: The metrics including UAR, spearman correlation, and pearson correlation in legal and car domains have slightly dropped, but the change is not significant. We have observed that the amount of data in these two domains is relatively small (around 5% to 6%). The model demonstrates good generalization for these two domains, with similar performance using in-domain and out-of-domain data. Additionally, the model trained has exhibited somewhat insufficient training. Therefore, adding or excluding training data of these two domains doesn't significantly impact the model's performance, and it's possible that some of the metrics for the in-domain performance are slightly lower.
>
> Thanks again for you valuable feedback.

---

### Official Review · Reviewer_uFZJ · 2023-08-04

**Typos Grammar Style And Presentation Improvements:** Line 030. conversational dialogues --…
**Soundness:** 3

**Excitement:**

3: Ambivalent: It has merits (e.g., it reports state-of-the-art results, the idea is nice), but there are key weaknesses (e.g., it describes incremental work), and it can significantly benefit from another round of revision. However, I won't object to accepting it if my co-reviewers champion it.

**Paper Topic And Main Contributions:**

This paper introduces a large end-to-end open-domain dialogue dataset with quality annotations for dialogue evaluation research. The dialogues are collected from a public question/answer platform, covering six different domains. The authors propose to annotate the collected dialogues into three different quality levels (0, 1, 2 for low/moderate/high qualities) based on six standard rules: grammaticality, relevance, consistency, empathy, proactivity and informativeness. In total, there are 100k dialogues are annotated. Results of multiple baseline models are provided to serve as baselines for future dialogue evaluation research.

**Questions For The Authors:**

1. For experiments on user sentiment influence, how do you know a dialogue is positive or negative? I do not see this type of annotation in
2. Does the collected dialogues not need any data cleaning process? If so, can you provide cleaning rules for this process?
4. In lines 449-453, have you tried different prompts for LLMs to score dialogues? Like Chain of Thought method to guide LLMs? It is very surprising that LLMs have such low performance.

**Reasons To Accept:**

The paper introduces an annotated open-domain dialogue dataset with 100k dialogues included, and the annotated dialogues are good for dialogue evaluation research.

**Reasons To Reject:**

1. The quality annotation process is confusing. There are six standard rules for annotators but only one annotation for each dialogue. For instance, grammaticality, relevance, consistency, and empathy are related to low quality. However, there is no annotation to know among the four rules, which one causes a dialogue to be annotated as low quality.
2. The quality annotation dose not contribute to generation-based method. If researchers want to evaluate generation ability of a dialogue model, there is no labels for evaluating grammaticality, relevance, consistency, empathy, proactivity and informativeness of generated responses.

**Reproducibility:**

3: Could reproduce the results with some difficulty. The settings of parameters are underspecified or subjectively determined; the training/evaluation data are not widely available.

**Reviewer Confidence:**

3: Pretty sure, but there's a chance I missed something. Although I have a good feel for this area in general, I did not carefully check the paper's details, e.g., the math, experimental design, or novelty.

---

> ### Author Rebuttal · Authors · 2023-08-29
>
> We sincerely thank you for your valuable feedback. The followings are our responses to your concerns in order.
>
> **Annotation with Reasons**: We annotate a holistic label fusing the six dimensions to align with human perspective for quality assessment. This mapping assigns distinct quality levels (e.g., 0, 1, 2) to the various dimensions. Moreover, each annotator is asked to provide explicit reasons when label (see Sections 4.2 and the Appendix A.2 for more details).
> There are two advantages of  providing explicit reasons: (1)  they help annotators to improve annotating precision and (2) they can be used to determine which specific standards that are unsatisfied for low quality. These explicit reasons might be more suitable for the evaluation on generation-based models.
>
> **Response to Question 1**: In the detailed analysis of whether user responses' sentiments would impact our quality assessment, we distinguish positive and negative sentiments via key words ("ok Thanks" or "I don’t think so"). We append this identified emotional expressions to the original dialogue content to validate any changes in quality assessment. We conduct this analysis to determine whether our model has learned shortcuts for quality assessment or not.
>
> **Response to Question 2**: Our data is randomly sampled from WenYiWen platform and we only collect those dialogues licensed by both the users and the respondents to be publicly published on the internet. We then sample dialogues from six domains: emotion, digital, legal, education, car, and finance. Finally, we filtered out dialogues containing privacy or personal information. This is achieved through our annotation process, where a score of -1 or -2 is assigned if there is privacy or personal information in the questions or responses. We subsequently filter out the dialogues with scores of -1 or -2.
>
> These descriptions are spread across Section 4.1, the ethics statement section, and Appendix A.2. We appreciate your feedback and we promise to provide a clearer of our data processing in Section 4.1 in the next revision.
>
> **Response to Question 3**: We utilized the explicit reasons and labels in the annotations for few chain-of-thought examples. We demonstrate a slight improvement in the quality assessment performance of the few-shot approach compared with the zero-shot approach.
> | Model | Acc | UAR | Kappa | Spearman | Pearson | Precision | Recall | F1 |
> | --- | --- | --- | --- | --- | --- | --- | --- | --- |
> | Ernie | 68.40 | 40.04 | 50.40 | 52.19 | 52.05 | 67.35 | 66.75 | 67.02 |
> | ChatGPT | 39.30 | 21.88 | 4.70 | 12.83 | 12.51 | 37.61 | 36.47 | 36.28 |
> | ChatGPT+CoT | 41.00 | 24.91 | 10.68 | 17.79 | 17.33 | 30.77 | 31.14 | 30.38 |
>
> Thanks again for providing these valuable feedback.

---

### Official Review · Reviewer_JK85 · 2023-08-05

**Soundness:** 4

**Excitement:**

4: Strong: This paper deepens the understanding of some phenomenon or lowers the barriers to an existing research direction.

**Justification For Ethical Concerns:**

N/A.

**Paper Topic And Main Contributions:**

This paper introduces a large-scale dialogue quality assessment dataset (DiQAD) that could be extremely valuable for the training and evaluation of future dialogue systems. The dialogues are annotated on six unique dimensions: grammaticality, relevance, consistency, empathy, proactivity, and informativeness. Furthermore, dialouages are from a wide variety of topics. Ultimately, the authors evaluate several state-of-the-art models on this dataset and provide a baseline for future studies.

**Questions For The Authors:**

OpenAssistant Conversations dataset (OASST1) is surprisingly absent in this paper. Is there a reason for this?

**Reasons To Accept:**

The amount of effort that has gone through developing this dataset is unmatched. This dataset is times larger than any other dataset in this domain. The authors claim users have to pay to talk to the experts; if true, I suspect the conversations are of high quality. Considering the conversations are in Chinese, this dataset is a valuable resource for future studies.

**Reasons To Reject:**

This paper could benefit from providing more details and clarity about the data collection process and the annotators. The Data Card and Human Evaluation Data Sheet are missing from this paper. Providing this information will facilitate the use and reproducibility of this work.

**Reproducibility:**

2: Would be hard pressed to reproduce the results. The contribution depends on data that are simply not available outside the author's institution or consortium; not enough details are provided.

**Reviewer Confidence:**

4: Quite sure. I tried to check the important points carefully. It's unlikely, though conceivable, that I missed something that should affect my ratings.

**Typos Grammar Style And Presentation Improvements:**

I found the paper slightly hard to read and understand, though I suspect this is due language barrier. I would suggest the authors have a native English speaker proofread the paper. For example, the language of the dataset should be mentioned earlier than on-page, line 345.

---

> ### Author Rebuttal · Authors · 2023-08-29
>
> We sincerely thank you for your valuable feedback. The following are our responses to your concerns in order.
>
> **Ethics**: As we mentioned in the ethics statement section, all the dialogues we collect have been authorized by the WenYiWen platform. We collect dialogues that are licensed by both the users and the respondents to be publicly available on the internet.
> Furthermore, annotators are asked to determine whether there are privacy and personal information present besides the quality labels (0, 1, 2). Dialogues with privacy concerns are excluded before release, as mentioned in the Appendix A.2.
>
> **Annotators**: The annotators are full-time employees who have continuously annotated the dialogue quality for WenYiWen for over a year, and all of them hold a college degree at least. They have undergone training to annotate the dialogue quality to meet the requirements before the formal annotation. Thus, we regard them as the experienced annotators. Moreover, the quality controllers are those annotators with higher annotation accuracy.
>
> The annotation team comprises 26 employees, including 14 females and 12 males. We promise to supply these information at the Appendix A.3 in the next revision.
>
> **Data Collection Process**: After we randomly sample the dialogues from WenYiWen platform that are licensed to be publicly available, we proceed to sample dialogues from six domains: emotion, digital, legal, education, car, and finance. Finally, we filtered out dialogues containing privacy or personal information. This is achieved through our annotation process, where a score of -1 or -2 is assigned if there are privacy or personal information in the questions or responses. We subsequently filter out the dialogues with scores of -1 or -2.
>
> **Response for Question 1**: Recently the open-assistants are constantly and rapidly improved and their response style are updated accordingly. The variety of open-assistants with their different machine generated styles, are different from real users settings. Therefore, we focus on the ultimate goal that enables machines generate human-epistemic high-quality responses. We directly annotated real human conversations for the dialogue quality assessment.
>
> Thanks again for providing these insightful feedback.

---

### Meta-Review · Area_Chair_kzPB · 2023-09-12

**Recommendation:** 4

**Metareview:**

The paper presents a large dataset (100k dialogues) for dialogue quality assessment; the dialogs are in Chinese and are annotated on six dimensions: grammaticality, relevance, consistency, empathy, proactivity and informativeness, and covers six different domains. The dataset is annotated manually despite its large size. The reviewers appreciate the analyses conducted on the dataset. The reviewers however point to additional work that should be cited and have questions about the annotation process and the annotators, to which the authors replied satisfactorily in the response (The annotators are full-time employees who have continuously annotated the dialogue quality for WenYiWen for over a year, and all of them hold a college degree at least. They have undergone training to annotate the dialogue quality to meet the requirements before the formal annotation. Thus, we regard them as the experienced annotators. Moreover, the quality controllers are those annotators with higher annotation accuracy. The annotation team comprises 26 employees, including 14 females and 12 males.)

---

### Decision · Program_Chairs · 2023-10-07

**Decision:**

Accept-Findings

**Comment:**

The paper presents a large dataset (100k dialogues) for dialogue quality assessment; the dialogs are in Chinese and are annotated on six dimensions: grammaticality, relevance, consistency, empathy, proactivity and informativeness, and covers six different domains. The dataset is annotated manually despite its large size. The reviewers appreciate the analyses conducted on the dataset. The reviewers however point to additional work that should be cited and have questions about the annotation process and the annotators, to which the authors replied satisfactorily in the response (The annotators are full-time employees who have continuously annotated the dialogue quality for WenYiWen for over a year, and all of them hold a college degree at least. They have undergone training to annotate the dialogue quality to meet the requirements before the formal annotation. Thus, we regard them as the experienced annotators. Moreover, the quality controllers are those annotators with higher annotation accuracy. The annotation team comprises 26 employees, including 14 females and 12 males.)